# OpenReview forum: "Exploring the Robustness of Decision-Level Through Adversarial Attacks on LLM-Based Embodied Models"
_acmmm.org/ACMMM/2024/Conference — MM2024 Poster_

### Official Review · Reviewer_qRHL · 2024-05-09

**Rating:** 3
**Confidence:** 3

**Summary:**

The paper explores the robustness of decision-making in large language model (LLM)-based embodied systems against adversarial attacks. It addresses a critical gap in the evaluation of LLM robustness in embodied scenarios by creating the Embodied Intelligent Robot Attack Dataset (EIRAD), which includes both targeted and untargeted attack scenarios. Two novel attack strategies are devised, alongside an attack success evaluation method using the BLIP2 model. Prompt suffix initialization techniques, tailored for specific tasks, are introduced to expedite the attack convergence process. The study also examines the effect of the number of initialized keywords on attack efficiency, revealing a positive correlation between the number of keywords and faster attack success. Comparative evaluations with established baselines like GCG and AutoDAN demonstrate improved attack success rates and efficiency with the proposed method.

**Strengths:**

Novelty: The paper introduces a first-of-its-kind exploration into the decision-level robustness of LLMs within embodied systems, filling a crucial research gap.
Technical Approach: The construction of EIRAD, a multi-modal dataset, provides a benchmark for future studies on LLM robustness. The introduction of attack strategies and the BLIP2-based evaluation method showcase technical innovation.
Adequate Evaluation: The work compares its methodology with existing attack algorithms, GCG and AutoDAN, demonstrating superiority in terms of attack success rate and efficiency. The use of ASR, ESR, and Epoch Cost as evaluation metrics adds rigor to the analysis.
Clarity and Applications: The paper is well-structured and clearly outlines the threat model and attack methodologies. The findings have implications for enhancing security protocols in embodied AI applications.

**Limitations:**

While the paper presents a valuable contribution, it might benefit from discussing potential countermeasures or defenses against the demonstrated attacks, providing a more comprehensive view of the robustness landscape.
There is limited discussion on the broader implications and scalability of the findings to different LLM architectures or real-world deployment scenarios.
The innovation of the method is limited and seems to be a simple application of previous jailbreak methods.

**Suitability:**

3

---

### Official Review · Reviewer_mmBR · 2024-05-24

**Rating:** 3
**Confidence:** 3

**Summary:**

This paper investigates the robustness of LLM-based embodied models against adversarial attacks, by introducing the Embodied Intelligent Robot Attack Dataset (EIRAD). The authors propose two attack strategies: untargeted attacks, which aim to generate outputs inconsistent with expected, and targeted attacks, which generate specific harmful outputs. Experimental results demonstrate the effectiveness of proposed methods in generating adversarial suffixes that mislead LLM-based embodied models, highlighting their vulnerability at the decision-making level.

**Strengths:**

- This paper provides a multi-modal dataset for the first time, which is specifically designed for evaluating the robustness of LLM-based embodied models and fills research gap.
- Experimental results show that the proposed method improve the ASR. Besides, the outputs have a strong correlation with the target task.

**Limitations:**

While the paper claims to propose a multi-modal dataset for exploring the robustness of decision-level on LLM-based embodied models, visual modality does not appear to play a role in the adversarial attack. Accoding to Figure 4, visual semantics are simply translated to text semantics through detection models. That is, adversarial attack is finished within only text-modal. So my question is, do we really need a LLM-based embodied model making decision in multi-modal scenario? In other words, all scene information provided by image can be replaced by text-generation, making multi-modal datasets less necessary.

**Suitability:**

3

---

### Official Review · Reviewer_hrEA · 2024-05-25

**Rating:** 3
**Confidence:** 3

**Summary:**

The paper investigates the vulnerabilities of embodied intelligent robots that integrate LLMs for decision-making. It highlights how attackers can manipulate LLMs to generate irrelevant or malicious outputs by altering prompts. To address this, the authors develop the EIRAD tailored for robustness evaluation and introduce two attack strategies: untargeted and targeted attacks. They also propose a novel method to assess attack success using the BLIP2 model, which considers the unique challenges posed by the integration of LLMs in embodied intelligence, thereby offering a comprehensive toolset for evaluating and enhancing the security of such systems.

**Strengths:**

1. The development of the EIRAD is a practical tool that fills a critical gap in testing the robustness of LLM-based systems in embodied environments.

2. The paper outlines both untargeted and targeted attack strategies, providing a broad spectrum for testing system vulnerabilities. Additionally, the novel method of using the BLIP2 model for evaluating attack success offers a more accurate measure of robustness tailored to embodied systems.

3. The study is well-documented and presented clearly, making the complex concepts accessible. The findings are directly applicable to enhancing the security of robotic systems, which is crucial for their safe operation in real-world scenarios.

**Limitations:**

1. The paper does not present a novel concept in evaluating the robustness of large models. There are existing studies that have explored similar themes, and a more thorough investigation and comparison with these studies would have added significant value.

2. The number of instances within the newly created EIRAD dataset might be too limited to provide a meaningful evaluation of robustness across diverse scenarios.

3. The evaluation of the proposed methods exclusively on the EIRAD dataset may limit the generalizability of the results. Including comparisons with other established datasets could have provided a broader validation of the findings.

4. While the paper discusses targeted and untargeted attacks, it does not fully explore the broader impacts of these attacks on system functionality and user safety. Discussing the potential real-world consequences of successful attacks could have underscored the practical importance of the research.

**Suitability:**

2

---

### Official Review · Reviewer_TNgk · 2024-05-25

**Rating:** 2
**Confidence:** 2

**Summary:**

This work introduces the Embodied Intelligent Robot Attack Dataset (EIRAD) to evaluate the robustness of LLM-based embodied agents. The proposed dataset includes two types of attack strategies: untargeted and targeted attacks.

In targeted attacks, the goal is to manipulate the embodied agent into executing a specific task (whether harmful or harmless) that deviates from its original goal. In untargeted attacks, the goal is to cause the agent to perform an action unrelated to its original objective, without any specific target in mind.

The authors propose a prompt-suffice initialization approach to devise these attacks, achieving a higher success rate compared to previous methods. Their key finding is that existing LLM-based embodied agents are vulnerable to such attacks, highlighting the need for further research on defense mechanisms to protect against these vulnerabilities.

**Strengths:**

Paper has done a fairly impressive evaluation with the existing works.

Existing work on this topic appears to focus solely on text-based embodied agents. This paper seems to be the first to propose an adversarial attack mechanism for multi-modal embodied agents or Vision Language Models (VLMs). However, I feel that this significant contribution needs to be more strongly emphasized in the presentation of their contributions

Proposed EIRAD dataset (if released open-source) will be a valuable source dataset for further research in this direction.

**Limitations:**

While I acknowledge the paper has done a commendable job in its evaluation and the proposed EIRAD dataset represents a significant contribution to the research field, I have several critical issues to highlight. These concerns have led to my decision of "Weak Reject." I look forward to the authors' rebuttal response to address these comments, which will help me make a more informed decision.

1. **Writing Quality:** A major issue with the paper is its writing. I strongly urge the authors to improve the writing to better communicate the motivation of the work, and particularly the methodology, which is currently quite unclear.

2. **Contributions:** The paper claims that "this work represents the first experiment in exploring the robustness of LLM-based embodied model decision-level processes." From my understanding, this is not accurate. GCG and AutoDAN were actually the first, and the proposed work seems to have evolved from these, particularly by addressing multi-modal information.

3. **Utilizing Multi-modal Information:** While the paper claims to be the first to evaluate the robustness of multi-modal embodied agents, it is unclear how their approach differs from GCG or AutoDAN. The key innovation and distinction of their approach, which purportedly leads to superior performance compared to these text-based embodied models, need to be clearly articulated.


4. **Prompt Suffix Initialization:** The key innovation in this work appears to be the initialization of the prompt suffix. However, it is unclear how the exact prompt suffix (which is an input to the LLM) is obtained to successfully achieve the attack. This aspect needs clearer explanation and elaboration.

5. **Optimization Equations:** Equations 1, 2, and 3 illustrate the optimizations used separately for both targeted and untargeted attacks. From my understanding, this optimization pertains to the response of the LLM for next token prediction. However, it is not clear how these equations help in determining the optimal prompt suffix that will be used as input to the model. The process of determining the prompt suffix needs to be clearly explained.

6. **ASR and ESR Distinction:** [Minor issue] The paper should also clearly distinguish between Attack Success Rate (ASR) and Execution Success Rate (ESR), as this distinction was unclear in the current version.

I look forward to the authors addressing these points in their rebuttal to provide clarity and strengthen the paper's contributions.

**Suitability:**

3

---

### Official Review · Reviewer_RYQr · 2024-05-28

**Rating:** 4
**Confidence:** 3

**Summary:**

This work represents the first experiment in exploring the robustness of LLM-based embodied model decision-level processes. A multi-modal dataset and applied method are established to evaluate the robustness in embodied scenarios. Extensive experiments show that the method improves attack success rate and attack efficiency.

**Strengths:**

1. It is essential to study the safety of LLM-based robotic models.
2. The scenario setup is intriguing, and the benchmark they proposed may be useful for the community of robotics safety.
3. The writing is good.

**Limitations:**

1. In Figure 2, the task-planning-based prompt for the Targeted Attack Data in the Harmful Attack scenario is oriented by the actual prompt rather than the target. According to the reviewer's understanding, it should be a task decomposition of the target.
2. The optimization of the prompt suffix is confusing. Is it related to prompt tuning [1]? If so, why not optimize the LLM model instead? Attacking the LLM model is more reasonable.
3. How is the multimodal dataset used? Is it based on the similarities of the decomposed tasks and the current image? Is a single image used instead of every frame captured during the robot's navigation and manipulation processes?


[1] Zhou, Kaiyang, et al. "Learning to prompt for vision-language models." International Journal of Computer Vision 130.9 (2022): 2337-2348.

**Suitability:**

2

---

### Meta-Review · Area_Chair_VDF2 · 2024-06-28

**Recommendation:** Accept (Poster)
**Confidence:** 5

**Metareview:**

The paper introduces the first experiment in exploring the robustness of LLM-based embodied models at the decision level, presenting a multi-modal dataset and methods to evaluate robustness in embodied scenarios. The proposed methods improve the attack success rate and attack efficiency, providing valuable insights for the community of robotics safety.
Extensive experiments show the effectiveness of the proposed method, improving attack success rates and efficiency compared to existing methods.
Despite some areas needing further clarification and improvement, the paper's novel exploration of the robustness of LLM-based embodied models at the decision level is a significant contribution. The proposed benchmark and methods are valuable for the field of robotics safety, and the comprehensive evaluation demonstrates the effectiveness of the approach.